# An open-source, low-cost voluntary running activity tracking tool for *in vivo* rodent studies

**Grace E. Deitzler**[1☉]*, **Nicholas P. Bira**[2☉], **Joseph R. Davidson**[2], **Maude M. David**[1,3]

**1** Department of Microbiology, Oregon State University, Corvallis, OR, United States of America, **2** Collaborative Robotics and Intelligent Systems (CoRIS) Institute, Oregon State University, Corvallis, OR, United States of America, **3** Department of Pharmaceutical Sciences, Oregon State University, Corvallis, OR, United States of America

☉ These authors contributed equally to this work.
* deitzleg@oregonstate.edu

## Abstract

*In vivo* rodent behavioral and physiological studies often benefit from measurement of general activity. However, many existing instruments necessary to track such activity are high in cost and invasive within home cages, some even requiring extensive separate cage systems, limiting their widespread use to collect data. We present here a low-cost open-source alternative that measures voluntary wheel running activity and allows for modulation and customization, along with a reproducible and easy to set-up code pipeline for setup and analysis in Arduino IDE and R. Our robust, non-invasive scalable voluntary running activity tracker utilizes readily accessible magnets, Hall effect sensors, and an Arduino microcontroller. Importantly, it can interface with existing rodent home cages and wheel equipment, thus eliminating the need to transfer the mice to an unfamiliar environment. The system was validated both for accuracy by a rotating motor used to simulate mouse behavior, and *in vivo*. Our recorded data is consistent with results found in the literature showing that the mice run between 3 to 16 kilometers per night, and accurately captures speed and distance traveled continuously on the wheel. Such data are critical for analysis of highly variable behavior in mouse models and allow for characterization of behavioral metrics such as general activity. This system provides a flexible, low-cost methodology, and minimizes the cost, infrastructure, and personnel required for tracking voluntary wheel activity.

## 1 Introduction

Voluntary wheel running behavior is a widely used metric to study motricity in rodent research. Laboratory mice will run voluntarily if they have access to a wheel in their cage [1]. Tracking the changes in voluntary running behavior can be used to evaluate behavioral changes (such as general activity levels which may correspond to anxiety, or adverse side effects from drug toxicity testing), exercise capacity (such as what might be observed in genetically modified or disease model mice) and physiological function. Running activity can be influenced by a variety of conditions including sex, age, diet, environmental conditions (ambient noise, temperature, etc) and strain of mouse. Most voluntary running activity occurs

**Data Availability Statement:** The list of materials, data, and all source code can be found at \url{https://github.com/MaudeDavidLab/Motricity_Tracker} \cite{Motricity2020}.

**Funding:** Research was supported by the National Science Foundation Graduate Research Fellowship under Grant No. 1840998 (NPB and GED), the National Institutes of Health Small Business Innovation Research Grant \#R44 DA043954 03 by NIH National Institute on Drug Abuse, and the Oregon State University College of Science Research and Innovation Seed (SciRIS-ii) Program award (MMD). The funders had no role in study design, data collection and analysis, decision to publish, or preparation of the manuscript. NSF GRFP: https://www.nsfgrfp.org/ SBIR: https://sbir.nih.gov/ College of Science at Oregon State University SciRIS-ii: https://internal.science.oregonstate.edu/rdu/internal-research-funding-program

**Competing interests:** MMD has financial interests relative to the activity of Second Genome, and Second Genome could benefit from the outcomes of this research. This does not alter our adherence to PLOS ONE policies on sharing data and materials. The other authors have no conflicts of interest to declare that are relevant to the content of this article.

during the dark phase of the 12 hr/12 hr light/dark cycle, because mice typically have nocturnal circadian rhythms [2].

Currently available systems for tracking voluntary running activity are often cost-prohibitive, sometimes requiring removing mice from the home cage during the tracking period, and several systems require the user to purchase expensive proprietary software licenses. Low-profile and minimally invasive wireless options on the market can cost several thousand dollars for a system that covers only 6 cages (what our proposed tool described here is designed for), with the bulk of that cost being for the licensure, wireless communication hubs, and computer hardware systems. Other systems can cost into the tens of thousands of USD for an entire housing system designed to replace the home cages. While these specialist home cage systems allow for collection of additional data (such as operant conditioning experiments), the cost of such systems can become a burden to researchers or instructors who are seeking basic activity tracking capabilities.

Mice are sensitive to changes in their environment and being transferred between unfamiliar cages can cause increases in levels of serum corticosterone and anxiety-like behaviors [3]. In order to accurately assess voluntary running behavior that is not influenced by external sources of stress, there is a need for a low-profile tracking system that keeps the mice in their native cage environment, with a wheel they are already familiar with. Existing systems, as mentioned above, either do not keep mice in their home cages or come with a price tag that includes cost-prohibitive software licenses and hardware that can make such systems inaccessible to smaller research groups or teaching facilities.

Here we outline a robust, system-agnostic, low-cost alternative tracking system that can be built in a lab or home setting with easily accessible off-the-shelf materials and open-source software. All documentation, an itemized list of materials, .STL files for optional 3D printing, and source code are provided on GitHub [4]. The proposed method involves the use of a small, non-invasive magnet that can be attached to the home cage wheel with minimal disturbance to the mice. While this method is specifically designed to be used on an angled disc style wheel, the system could easily be adapted to other wheel types, as long as the magnet can be securely affixed. This system is ideal for laboratories that would like to measure the voluntary wheel activity in smaller-scale experiments that do not require the use of hundreds of cages. We describe the use of this system for six sensors, but it could easily be scaled up to 18 sensors with a single Arduino Uno, or more with different models of microcontroller or additional microcontrollers per computer. As long as the laptop computer is powered on, the system can run indefinitely, allowing for tracking over multiple nights or even weeks. In the following sections, we discuss how the system was built and validated, and then analyze data collected with the system in two usage scenarios.

## 2 Materials and methods

### 2.1 Building the tracker system

The tracker system design is based on the principles of a rotary encoder, a device frequently utilized within robotics, automotives, and other industries. Its key function is to sense the state of a rotating shaft and provide feedback on the shaft's angular position, speed, and/or direction of rotation. Rotary encoders frequently make use of Hall effect sensors and small magnets to encode this data, tracking the rotation of a shaft by triggering the sensor each time a magnet passes nearby.

For this application, the intention was to enable tracking of multiple mice in a single cage, with minimal changes to their known environment. This need for minimal invasion, as well as the need to avoid wires in the cage that could pose a danger to the mice, suggested a wireless

solution was necessary. To this end, slightly altering the existing exercise wheel to attach a small magnet, which can then wirelessly trigger a Hall effect sensor attached to the exterior of the cage, achieves these goals. This method allows for a scalable approach to multiple cages with mouse wheels, and allows for simultaneous and continuous monitoring of mouse activities levels, without significantly altering the cage environment.

For the tracking system, an Arduino Uno R3 microcontroller was selected due to its ease of programming and component integration, low cost, and wide availability of numerous open-source software libraries. Normally, an Arduino Uno only has two pins available for interrupts. Interrupts are necessary for real-time monitoring, as the triggering of the hall effect sensor can happen at any time. The Arduino code was written with this in mind, and made use of the EnableInterrupt library [5]. This additional library enables assigning interrupt functionality to all pins on the Arduino, allowing for multiple sensors beyond the default two. The Arduino was connected to a laptop over USB, and generates a .txt file containing the timestamps and distance traveled through communication with the software PuTTY running on the laptop. Beyond the average velocity and distance traveled discussed in this paper, this data can be analyzed to measure peak velocities, distribution of periods of rest, and other useful metrics for mouse activity.

**2.1.1 Assembling the wheel and cage.** Wheels used were angled running discs on plastic huts (Bio-Serv, product numbers K3328 and K3251 (Fig 1A) with a diameter of approximately 15-20 cm, made of solid plastic flooring, attached to the top of the plastic hut. A small magnet, 11 mm in diameter, was affixed to the outer edge of the wheel using either an epoxy or placed into a small 3D printed jacket slid over the edge, held in place with a small amount of non-toxic PVA glue. The wheel was then reattached to the hut and aligned with the front side of the cage in such a way that the flat face of the magnet passed by the wall in its trajectory. A small amount of non-toxic PVA glue was applied to the bottom of the hut and let dry so that the hut would not move during the testing period. Following the alignment of the wheel to the wall, the Hall effect sensor was affixed to the outside of the wall of the cage adjacent to the path of the magnet during a full wheel rotation (Fig 1B). Mice were placed in the home cages after the setup was complete.

# 3 Bill of materials

Assuming access to 3D printing (many universities provide 3D printing services at low cost to students and researchers), preexisting rodent cages and wheels, and ownership of a laptop computer, the total cost for implementing the designs described above totals approximately $70 USD at the time of this paper's submission, as shown in Table 1 (see Table 2 for a more detailed description of each component). A laptop computer capable of running PuTTY or a similar program for serial communication with the microcontroller is required. A Prusa MK3S desktop 3D printer was utilized to fabricate the 3D models for custom housings for both the magnets to attach to the wheel and the electronics. This step is optional, but does make the removal of the magnet for cleaning simpler. Making use of cheaper components, or sourcing from other retailers, could drop the total price below $50 USD as of October 2021.

## 3.1 Validating the tracker system: Mechanical test

A robotic "mouse" was created to test the functionality of the testing apparatus. This "mouse" consisted of a single foam wheel attached to a DC motor and Arduino to be rotated at a constant velocity for three intervals. This foam wheel was placed upon one of the mouse wheels with an attached magnet, and five sensors were placed adjacent to each other, following the curvature of the rotating mouse wheel. The system recorded for 85 seconds at a sampling

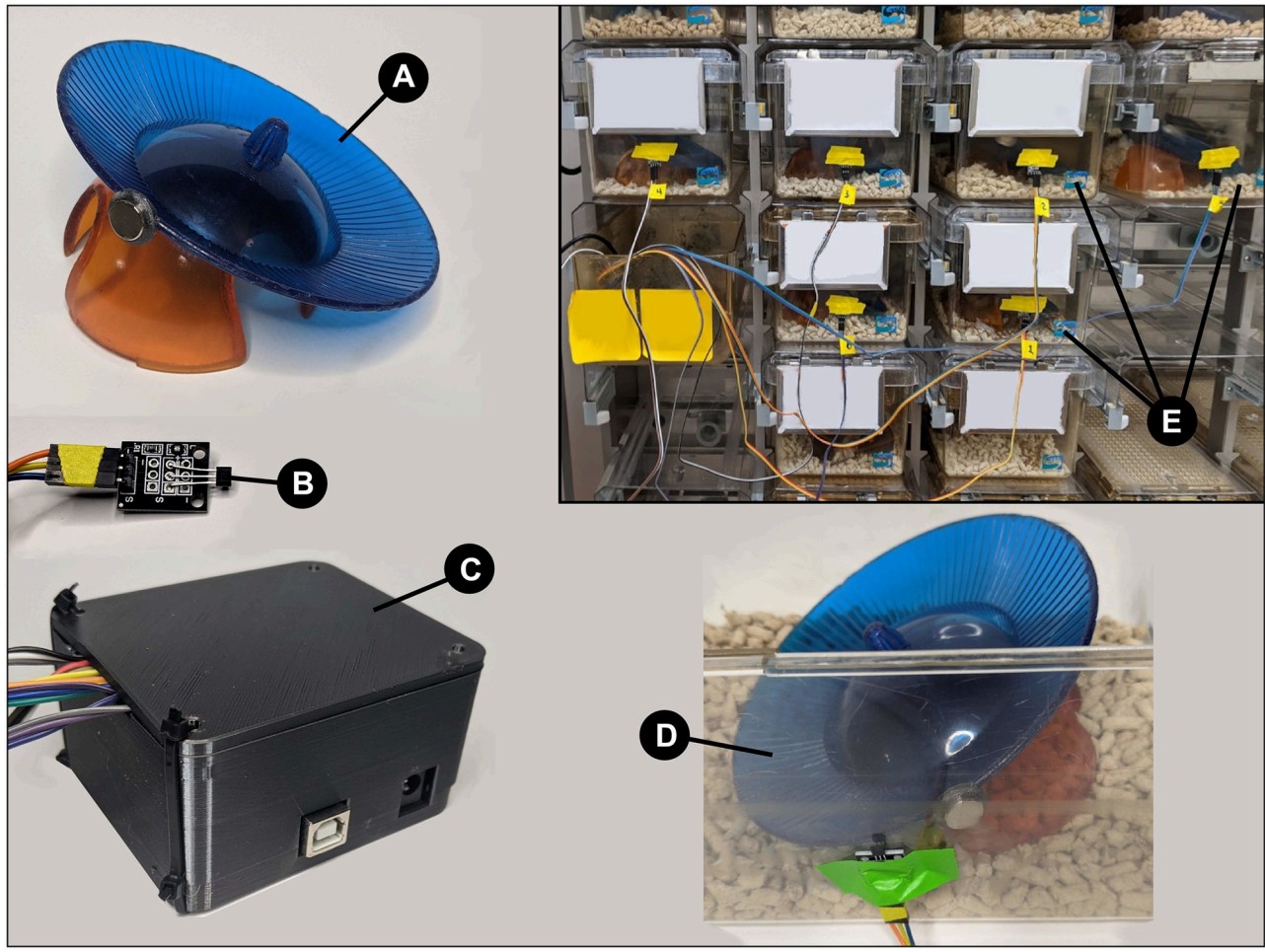

**Fig 1.** A. The magnet attached to the wheel and hut. B. TheHall effect sensor, connected via wires to the Arduino. A red LED indicates when the sensor detects a magnet. C. The 3D-printed housing for the Arduino and breadboard, with wires for six sensors leaving the enclosure. D. Aligning and connecting the Hall effect sensor to overlap the rotation path of the magnet inside the cage. E. Example of the system connected to 6 cages. The wires are all leading back to the Arduino, which is connected via USB to a laptop.

**Table 1. A bill of materials for constructing the tracker system.**

| Part | Cost |
|---|---|
| A) Arduino Uno (R3) | $23.00 |
| B) Hall Effect Magnetic Sensors (A3144) | $6.00 |
| C) Small Magnets (10 mm x 3 mm) | $14.00 |
| D) Non-toxic PVA Glue | $8.26 |
| E) Long Breadboard Jumper Wires | $9.00 |
| F) Breadboard | $9.00 |
| G) 3D Printing Filament (PLA, 80g) | $2.40 |
| Total: | $71.66 |

**Table 2. Detailed description of each component.**

| Part | Purpose | Cost (in USD at time of Submission) |
|---|---|---|
| A) Arduino Uno (R3) | An open-source, low-cost microcontroller for the purpose of communicating with all mouse wheel sensors and recording the ongoing distance traveled. | $23.00 |
| B) Hall Effect Magnetic Sensors (A3144) | Small, standardized magnetic sensors to detect the state of a magnetic field. These include the necessary pull-up resistor and a red LED on an integrated PCB, simplifying circuit construction and allowing for visual validation of sensor triggering. | $6.00 |
| C) Small Magnets (10mmx3mm) | Once attached to the mouse wheel, the magnet rotates around the perimeter and triggers the hall effect sensors to count a single wheel rotation. | $14.00 |
| D) Non-toxic PVA Glue | Washable adhesive necessary to glue the wheel in place, ensuring proper alignment between the sensors and the rotating magnet. | $8.26 |
| E) Long Breadboard Jumper Wires | Wires to connect the hall effect sensors to the microcontroller. | $9.00 |
| F) Breadboard | Hub for wiring to break out shared power and ground for all sensors. | $9.00 |
| G) 3D Printing Filament | A 1 kg roll of PLA filament costs around $30 and the printed components of our system utilize approximately 80g to print. | $2.40 |
| Total: | - | $71.66 |

frequency of 4 Hz, and recorded the data shown in Fig 2. This validation demonstrated 100 percent agreement between all sensors at the end of the test, and any variability present during the recording process arose as an artifact of the PuTTY sampling rate and not the sensor reliability.

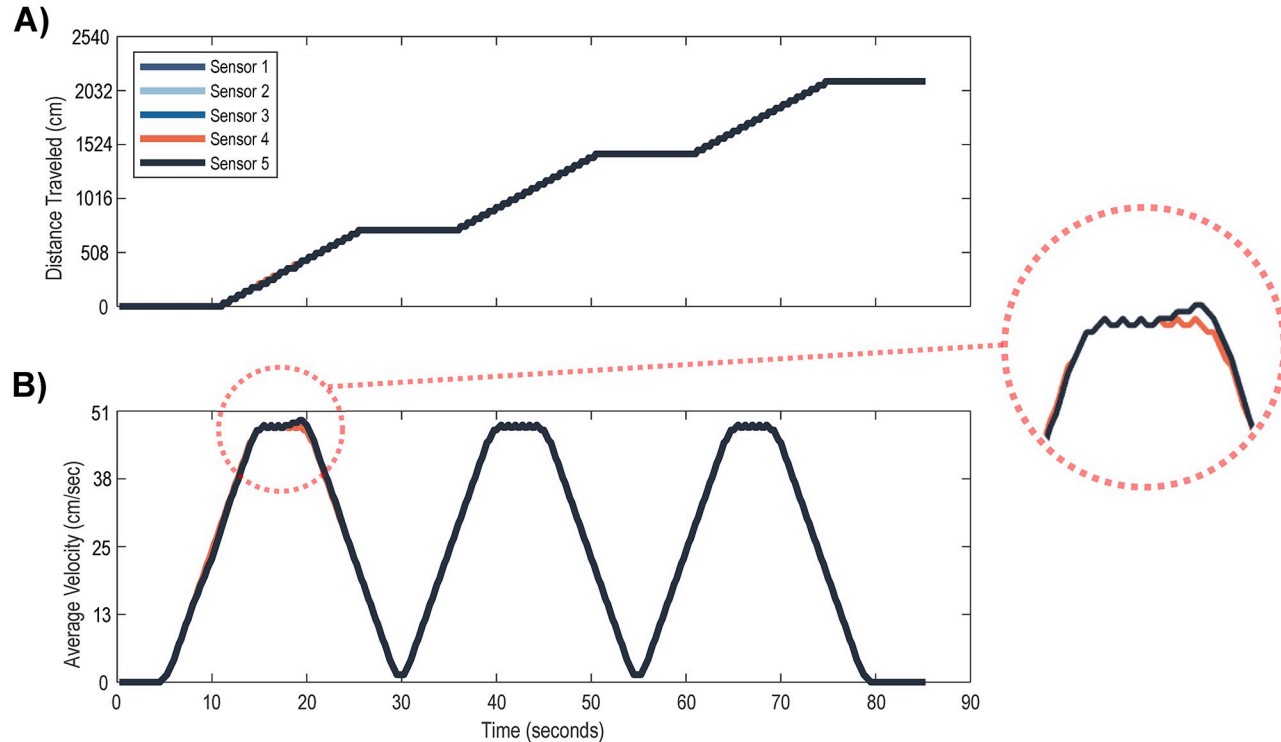

**Fig 2.** A. Top: The measured distance traveled by the mouse wheel over the course of 85 seconds for five sensors at once. B. Bottom: Average velocity of the mouse wheel with a small (4) averaging window.

### 3.2 Validation experiments: *In vivo*

All mice in this validation test were retired breeder C57BL/6J females (Jackson Laboratories) and were approximately 15 weeks of age. Mice were excluded if they were in poor health or had previously received experimental treatments from other ongoing laboratory projects. Validation of the tracker system was accomplished by two means: assessing the distance and velocity by four mice in a single cage across two nights ($n = 1$ cage), and assessing the distance traveled by the same four mice split into four separate cages across two nights ($n = 4$ cages). Mice remained in their home cage in the facility for the first part of the test, and were moved into new individual cages for the second part. A wheel with the magnet attached replaced their previous wheel (same style and brand, sans magnet.) Tracking began at 7 PM after the dark cycle at the mouse facility had begun and ended at 8 AM on the morning of the second day. The tracking therefore occurred over an entire light-dark cycle and an additional dark cycle. During this time mice had typical ad libitum access to their standard chow and water. To reduce potential for confounding effects, the same handler was responsible for all setup of tracker system and any handling of mice, and all studies were carried out in the same room in the mouse facility. All procedures and experiments involving mice performed in the study were carried out according to and approved by the Oregon State University Institutional Animal Care and Use Committee, on Animal Care and Use Protocol #5127.

### 3.3 *In vivo* usage scenario experiment comparing wheel activity prior to and during caloric restriction

All mice used in this experiment were male, between 10-13 weeks of age at the start of the test. Mice were CNTNAP2 knockout strains (B6.129(Cg)-Cntnap2$^{tm1Pele}$/J, Homozygous genotype, from Jackson Laboratories, JAX stock number 017482) [6]. Each of 8 cages contained three mice ($n = 8$ cages) and were available to the authors due to a concurrent study occurring in the laboratory. Mice were fed a normal chow diet from weaning, before 80% caloric restriction for three days. Running activity was measured for approximately 10 hours overnight prior to restriction to establish a baseline, and for 10 hours overnight following the three-day period of caloric restriction. All procedures and experiments involving mice performed in the study were carried out according to and approved by the Oregon State University Institutional Animal Care and Use Committee, on Animal Care and Use Protocol #5127.

### 3.4 Data processing and statistical analysis

Statistics, processing, and plotting for in vivo experiments were done using R version 3.6.3 (R Core Team, 2020), the ggplot2 (v3.3.2) [7], and reshape2 (v1.4.4) [8] packages. Processing and analysis of the "robotic mouse" data was done in MATLAB (R2020a). The full reproducible code and data can be found on our GitHub [4]. To test our hypothesis for the caloric restriction test, we used a permutation test on the means of the differences between each cage before/after caloric restriction distance traveled. In more detail: we measured the distance each cage ($n = 8$, with 3 mice per cage) ran for 3 days, 10 hours a day and summed that distance per cage ('before distance'). Then we measured the distance each cage ran after caloric restriction for 3 days, 10 hours a day, and summed that distance ('after distance'). We generated 1000 permutations by randomly shuffling each cages' before/after distance, and took the mean across cages. These means produce the null distribution. We then measured the actual mean of before/after distances, and calculated the area under the curve that is more extreme than the actual measured mean value. We chose this test over a t.test to account for the non-normal distribution,

and over a Wilcoxon test as we wished to test the difference in the means in a non-parametric way (rather than the rank sum).

## 4 Results

### 4.1 Validating the tracker system: Mechanical test

While the robotic mouse was active, the apparatus was rotated at a known constant speed and traveled a set distance before pausing (Fig 2A). This was controlled programmatically while the number of rotations were visually observed as a control. All five sensors recorded the wheel rotations simultaneously, resulting in the plot below where each sensor output is over-laid. The only slight variation in recording between sensors is visible during the first interval (Fig 2B near the 20 second mark, highlighted in red on the right side of the figure), when the rotation of the wheel matched up with the refresh rate of the Arduino (0.25 seconds) resulting in a slight misalignment between record keeping at that specific time point. However, this mis-alignment disappeared after a few more rotations, resulting in equivalent record keeping for all sensors and demonstrating the robustness of the system for collecting wheel rotations simultaneously. In Fig 2B, the same distance data was plotted as an average speed with a mov-ing average of 4 data points to smooth the plotted line. This visualization shows the top speed of the mouse wheel to be around 44.5 cm/sec while the robotic mouse was moving, and 0 while not moving. The recorded data was in complete agreement with the distance traveled by the robotic mouse both from visual counting and programmed distance traveled.

### 4.2 Validation experiments: *In vivo*

When four mice were in a single cage, the total distance run over 45 hours (indicated in blue) was just under 32 kilometers (Fig 3A). However when mice were separated into four different cages, each individual mouse ran between 2 and 12 kilometers, altogether totaling less than 24 kilometers (Fig 3B). Fig 3A also shows the change in velocity over the course of the study, reflecting changes in activity as the light-dark cycle changes. This test shows that our system can accurately pick up the changes in rate over time over the two days, and when considering the light-dark cycle can reflect the changes in the rodents' active hours.

### 4.3 *In vivo* caloric restriction in CNTNAP2 mice

Here we hypothesized that restricting the caloric intake of mice would result in an increase in locomotor activity as measured by distance traveled, compared to the baseline of typical caloric intake. An increase in distance traveled was observed in most cages following caloric restric-tion (see Fig 4). This difference in distance between the baseline and caloric restriction mea-surements was not found to be statistically significant, but did show an upward trend. To test our hypothesis we calculated the null distribution using 1,000 label permutations and found the area under the curve of the null distribution to assess significance (p = 0.0876). On the night of tracking following caloric restriction, bedding was stuck in Cage 5's wheel and thus no data was collected, so Cage 5 was removed from the statistical analysis.

## 5 Discussion

Our aim to create a low-cost and accessible testing apparatus which implements open-source code pipelines and provides an affordable option to laboratories with minimal resources and personnel was successful. Here we have presented two *in vivo* usage scenarios to demonstrate the efficacy of this low-cost system. We were able to track distance and velocity over the course of the experimental period the mice ran on their wheels. Fig 3A demonstrates that the mice are

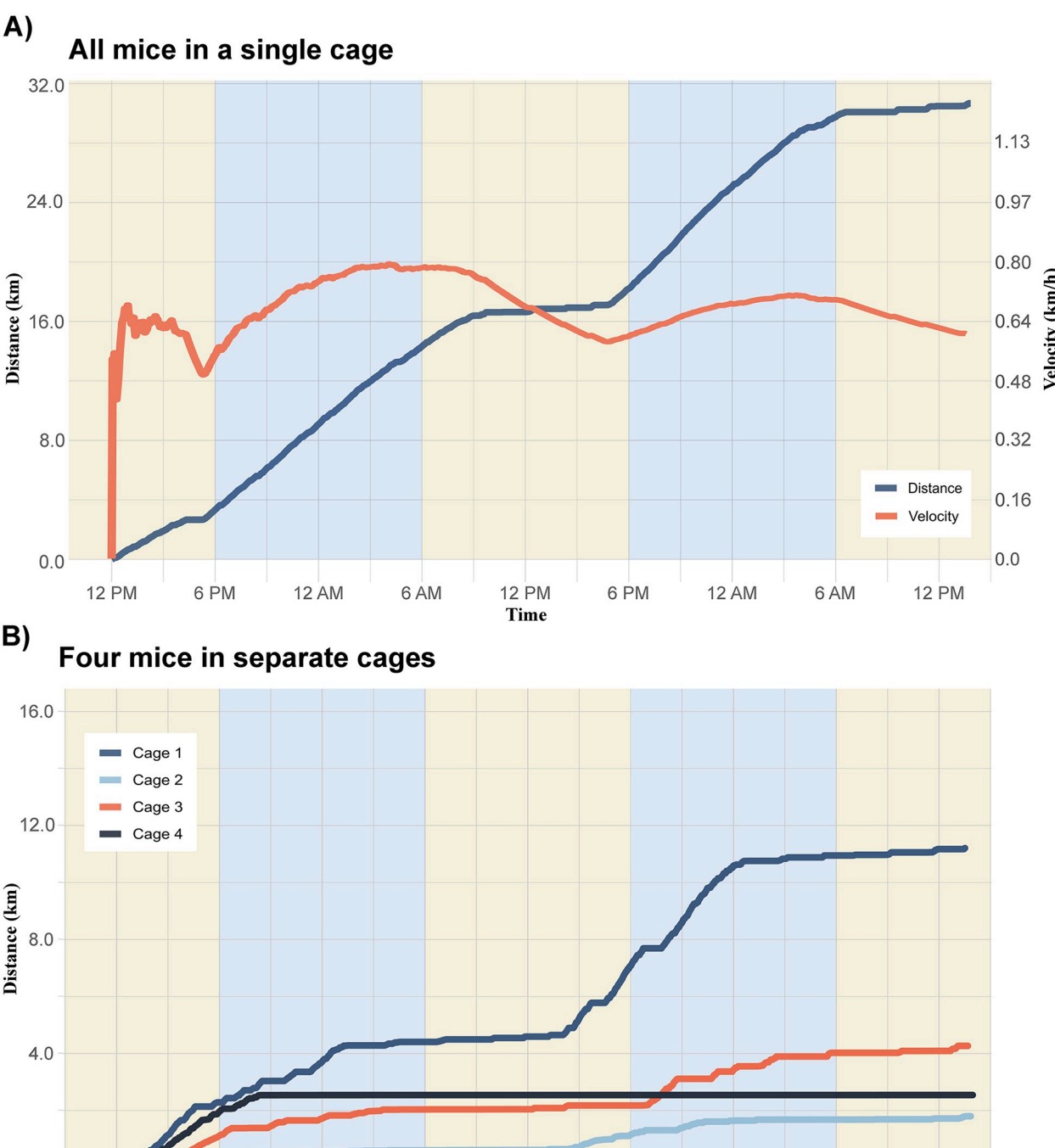

**Fig 3.** A. The measured distance traveled (blue) and velocity (red) by a single mouse wheel over the course of 50 hours for four mice in one cage. Yellow and blue panels indicate the light-dark cycle throughout the study period B. Measured distance traveled for four mice in individual cages over the course of 50 hours. Upon inspection during take down of the system, Cage 4 appeared to have inhibited the wheel with bedding at some point during the first night, so rotations of the wheel were not possible, and thus the tracker ceased measurements.

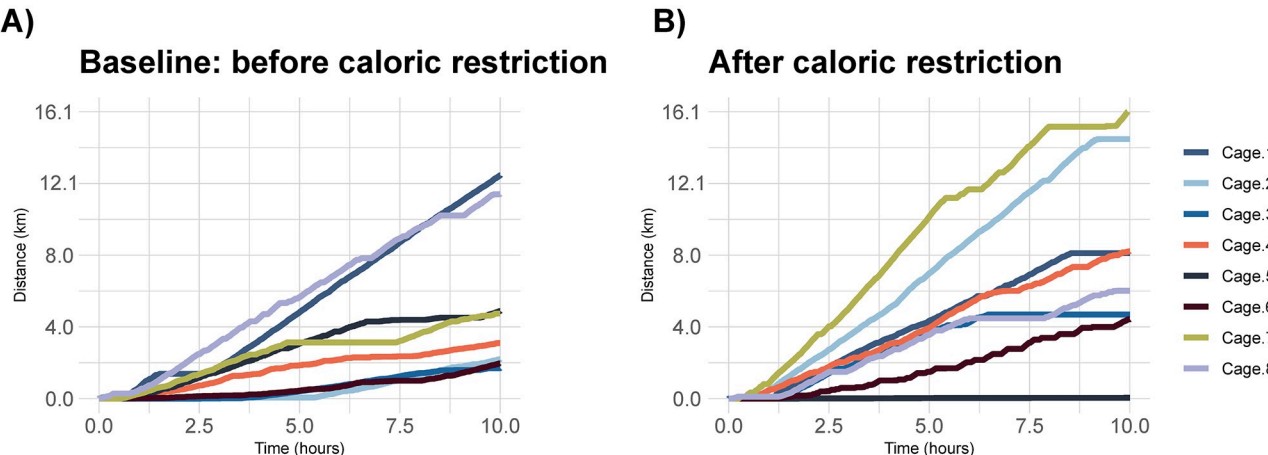

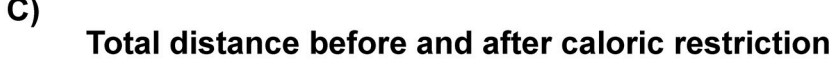

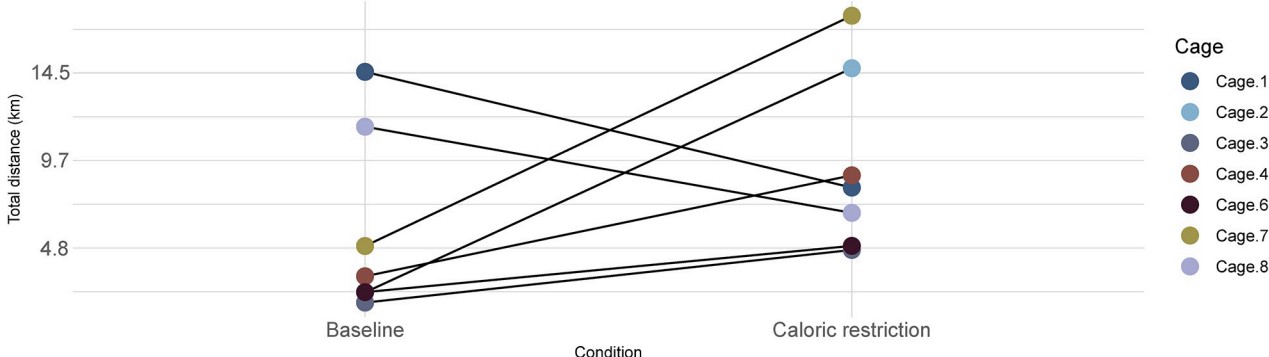

**Fig 4.** A. Distance traveled overnight by each cage prior to caloric restriction. B. Distance traveled overnight by the same cages following three days of caloric restriction at 80% of the normal diet. C. Comparison of total distance before and after caloric restriction. When Cage 5 was removed from the analysis, the difference between baseline and caloric restriction was not found to be statistically significant (permutation test, p = 0.0876).

most active during the dark hours, and that the changes in velocity correspond with the change in the slope of the distance. We were also able to detect differences in the cages depending on whether there was a solitary mouse or a group of mice. This disparity could be explained by the fact that as social animals, mice tend to be more active when paired in cages with other mice. On their own they may not run as much as when they are all together. Additionally, a single more active mouse could be driving the total distance in the cage together. Finally, for the portion of the study assessing mice in individual cages, mice were moved out of the home cage they had resided in prior to the study, and thus this could have introduced stress, affecting their activity levels.

Although we did not find a significant difference in distance traveled before and after the caloric restriction period of the CNTNAP2 mice, we did observe a trend indicating mice traveled more following the restriction period. Despite the reduction in caloric intake, an increase in activity (here measured by distance traveled over the night) has been observed in several rodent studies [9]. The hypothesized explanation for this phenomenon is that an increase in foraging activity during times of nutritional stress is required for survival by rodents in the

wild, so the general upward (though not significant) trend in wheel running which we observed here could be interpreted as a proxy for that foraging and searching behavior. A future study to robustly assess these changes using our described tracker system would need to include a much larger number of cages in order to have sufficient power to detect a truly significant effect.

The proposed system is ideal for short-term (less than a week) monitoring of wheel running as a measure of general activity in assessments such as post-surgery monitoring or short-term drug toxicity studies, but could also be utilized for long-term observation of patterns or behavioral responses to stimuli. Short-term experiments are optimal with the described system in that eventually the amount of recorded data could become large and difficult to computationally manage during analysis, and longer experiments have a greater chance of mechanical disruption (bedding jamming the wheel), but this can be circumvented by checking the cages and saving ongoing files to the computer every few days. The system only disturbs the home cage during the setup, and can subsequently record data for as long as necessary without removing or altering the home cage at all. Additionally, personnel is only needed during setup and resets (as described in the scenario above for running longer-term measurements), as the system can run consecutively as long as the laptop it is connected to remains powered. A limitation of this approach is that cage cleaning and feedings may require realigning the sensors and wheels, and result in a restart of data collection. To improve this alignment, making slight modifications to the cages and wheels by adding pegs and small holes would ensure reliable wheel and sensor alignment without manual adjustment and gluing. Our results show that the system can produce results that align with the current literature on how far mice run in a 24-hour cycle [2], and our validation via controlled rotation of the wheel demonstrates that our system can accurately measure wheel rotations. The tracker can run during both light and dark cycles in the facility. This gives researchers the advantage of observing the full range of wheel activity, including nocturnal behaviors.

Our system also presents some limitations. One such limitation is that with multiple mice in a single cage our system cannot detect which mouse is running (or how many mice are running, total) on the wheel at any given time, as shown in Fig 3A. For this reason, many cage-compatible tracking systems traditionally house mice individually. However, single-housing of rodents can lead to increased anxiety, a reduction in cognitive performance, and increased biological stress markers [10]. Such issues could be resolved with the use of the system in conjunction with a video camera outside the cage, along with video tracking and RFID tools that can identify the number of animals on the wheel at any given time [11]. Testing systems such as the one developed by Singh et al [12] achieve similar goals with a visual tracking system, and combining both the system presented here with the capabilities demonstrated in this study would further deepen the richness of collected data. Furthermore, we only present here a setup allowing tracking of six cages at a time. However, Arduino-based microcontrollers used with the EnableInterrupt Library enable up to as many inputs as there are available pins on the microcontroller. This allows scalability for researchers wishing to implement the system in additional cages, for up to 18 for an Arduino Uno and even more for larger microcontrollers such as the Arduino Mega (up to 54). A single laptop computer could accommodate multiple microcontrollers at once through USB, enabling multiple microcontrollers recording multiple cages in parallel, should a researcher wish to scale the system beyond what a single microcontroller can record. Another limitation of this wheel-sensor design is that it utilizes a single sensor. Our counting methodology assumes one reading on the sensor is equivalent to one rotation of the wheel. This holds true for continuous use, but may introduce error in the long term as mice get on and off the wheel, resulting in partial rotations. One way to validate full counts would be to place another sensor on the opposite side of the wheel, and only count a

rotation when both sensors trigger in sequence; however, this would double the number of sensors, while being difficult to implement without bringing the sensors fully into the cage, undermining the overall value of the system. As a result, one sensor was deemed sufficient for our tests, and cumulative error from miscounting artefacts are likely multiple orders of magnitude lower than the total distance traveled overnight (within a few centimeters accounting for when mice get on or off the wheel.)

## 6 Conclusions

Studying complex animal behavior such as voluntary wheel running often involves moving the mice from their home cage environment into an unfamiliar apparatus, requires extensive time and personnel to set up, and is costly. The stress of placing mice in an unfamiliar environment may cause spurious phenotypic results, possibly influencing behavior and metabolism of murine subjects. Monitoring behavior in the home cages with minimal alterations to the wheel or cage therefore confers a great advantage to the researchers wishing to study voluntary running behavior.

In this paper we present a low-cost, open-source, minimally invasive and reliable voluntary wheel running tracking system that can be scaled and implemented in the home cage of the rodents. We demonstrated that this system provides reliable and robust tracking capabilities and is low-cost with accessible, off-the-shelf materials. The open source nature of the system allows for expansion in both the hardware and the software, leaving open the possibility for such modifications as setting automatic timers for data collection, automated uploading of the data to an online repository, and expanding the number of cages run at a single time. The system is able to be scaled up for high-throughput analyses, and is suitable for remote running activity monitoring in the home cage as a useful tool for behavioral analysis.

## Acknowledgments

The authors would like to thank Mae Araki and Dr. Jennifer Sargent at the Laboratory Animal Resource Center at Oregon State University for animal care support and consultation. We would also like to thank Alexandra Phillips and Maya Livni for their assistance in tracker setup for the cages in the caloric restriction study, and Christine Tataru for guidance on the statistical analysis.

## Author Contributions

**Conceptualization:** Grace E. Deitzler, Nicholas P. Bira, Maude M. David.

**Data curation:** Grace E. Deitzler.

**Formal analysis:** Grace E. Deitzler, Nicholas P. Bira.

**Funding acquisition:** Maude M. David.

**Investigation:** Grace E. Deitzler, Nicholas P. Bira.

**Methodology:** Grace E. Deitzler, Nicholas P. Bira.

**Project administration:** Grace E. Deitzler.

**Resources:** Nicholas P. Bira.

**Software:** Nicholas P. Bira.

**Supervision:** Joseph R. Davidson, Maude M. David.

**Validation:** Grace E. Deitzler, Nicholas P. Bira.

**Visualization:** Grace E. Deitzler, Nicholas P. Bira.

**Writing – original draft:** Grace E. Deitzler, Nicholas P. Bira.

**Writing – review & editing:** Grace E. Deitzler, Nicholas P. Bira, Joseph R. Davidson, Maude M. David.

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
