## [Decision Letter · Decision Letter 0]

15 Jun 2022

PONE-D-21-39460An open-source, low-cost voluntary running activity tracking tool for in vivo rodent studiesPLOS ONE

Dear Dr. Deitzler,

Thank you for submitting your manuscript to PLOS ONE. After careful consideration, we feel that it has merit but does not fully meet PLOS ONE’s publication criteria as it currently stands. Therefore, we invite you to submit a revised version of the manuscript that addresses the points raised during the review process.

We look forward to receiving your revised manuscript.

Kind regards,

Dragan Hrncic

Academic Editor

PLOS ONE

Journal Requirements:

Research was supported by the National Science Foundation Graduate Research Fellowship under Grant No. 1840998 (NPB and GED) and the National Institutes of Health Small Business Innovation Research Grant \\#R44 DA043954 03 by NIH National Institute on Drug Abuse (MMD).

NSF GRFP: https://www.nsfgrfp.org/

SBIR: https://sbir.nih.gov/

Research was supported by the National Science Foundation Graduate Research 262

Fellowship under Grant No. 1840998 (NPB and GED) and the National Institutes of 263

Health Small Business Innovation Research Grant #R44 DA043954 03 by NIH National 264

Institute on Drug Abuse (MMD).

However, funding information should not appear in the Acknowledgments section or other areas of your manuscript. We will only publish funding information present in the Funding Statement section of the online submission form. 

Research was supported by the National Science Foundation Graduate Research Fellowship under Grant No. 1840998 (NPB and GED) and the National Institutes of Health Small Business Innovation Research Grant \\#R44 DA043954 03 by NIH National Institute on Drug Abuse (MMD).

NSF GRFP: https://www.nsfgrfp.org/

SBIR: https://sbir.nih.gov/

MMD has financial interests relative to the activity of Second Genome, and Second Genome could benefit from the outcomes of this research. The other authors have no conflicts of interest to declare that are relevant to the content of this article.

Reviewers' comments:

Reviewer's Responses to Questions

**Comments to the Author**

1. Is the manuscript technically sound, and do the data support the conclusions?

Reviewer #1: Partly

2. Has the statistical analysis been performed appropriately and rigorously? 

Reviewer #1: No

3. Have the authors made all data underlying the findings in their manuscript fully available?

Reviewer #1: Yes

4. Is the manuscript presented in an intelligible fashion and written in standard English?

Reviewer #1: Yes

5. Review Comments to the Author

Reviewer #1: In their manuscript „An open-source, low-cost voluntary running activity tracking tool for in vivo rodent studies “, the authors describe a “simple” solution to a cost-effective method for tracking activity data in a multi-cage setup. They use an Arduino microcontroller to process the data and Matlab/R to evaluate the sensor robustness and statistics. An in vivo caloric restriction experiment is selected as a use case to demonstrate the method.

First, I would like to say that I am very familiar with commercial systems that can do the same but are rather pricy. Therefore, I welcome this “hands-on” method to establish a validated and low-cost solution to obtain multiple time-resolved activity patterns. Furthermore, getting total wheel-running counts over time (e.g., overnight) is trivial. Therefore, the innovative focus of this study lies in a) cost-effectiveness, b) the ability to run n cages in parallel, and c) the time resolution of activity patterns. I find it a bit sad that the analytical focus of the analysis was not more on c), but I also understand that the main aim of the manuscript was to present the validated setup.

The points a-c are addressed in the manuscript. I want to thank the authors for their work and encourage them to continue this kind of work and, e.g., develop more measures for severity assessment that can be implanted this easily.

The manuscript is well-written, good to understand, and straightforward. The authors also address critical behavior-related activity issues like housing conditions, increased anxiety, and food-related stress.

However, some minor things need to be addressed before I can recommend the manuscript for publication.

a) Since this is an animal study (at least the caloric restriction experiment): did the authors include the ARRIVE guidelines with the manuscript?

b) In the caloric restriction experiment, there is no hypothesis. However, the authors mention that they measured the difference in distance between “baseline and caloric restriction measurements” and that this was statistically significant.

o I cannot see an initial hypothesis and how the effect is (potentially) biologically meaningful

o I guess that there was no power analysis done before the experiment. Therefore, we cannot know whether the result is sufficiently powered or meaningful. If the authors have done an a priori power analysis, they should include it (with their hypothesis).

If they haven’t, they should explain their level of significance threshold.

o The term “statistically significant” is not self-sufficient as a result. Without an effect or hypothesis, this statement is meaningless.

o A type-1 error or α-error of p<= 0.05 is the threshold for “statistical significance” in general science. However, the results report “p=0.0876” (line 170) above this threshold. Therefore, the result is NOT significant. Why was it termed “significant” in the text?

o However, scientists can (when there is a good reason) change that threshold, e.g., to p=0.1 (e.g., if they followed Fisher’s definition of the p-value). In this case, the result would be significant. But without a hypothesis, this is again meaningless, and the authors also give no reason why the level should be larger than the commonly accepted threshold.

o Please provide context for the reported p-value and why this should be significant and/or adjust the reporting of the result.

c) Typo in line 170: “varFiation” (variation?)

d) The authors analyzed the data “through calculation of the empirical cumulative distribution using 1,000 label permutations, followed by a one sided tailed test”.

o This needs a better explanation: What kind of test did the authors use and why?

o I understand a permutation test and that this test can, e.g., be a t-test that is permuted on the ECDF data differences. Was this a t-test?

o And, why was a one-tailed test chosen and not a two-tailed (this information should be given in the missing hypothesis as mentioned above)? The one-tailed design hints that the authors at least expected a lower/higher development in one of the groups, other than a general difference (two-tailed).

o ECDF functions can also be analyzed with a Kolmogorov-Smirnov test; Was it a KS-test?

o The term “one sided tailed test“ is incorrect. Usually, this is called a one-tailed xy-test. Sided and tailed means the same here.

In light of these points, I suggest a minor revision before publication.

6. PLOS authors have the option to publish the peer review history of their article (what does this mean?). If published, this will include your full peer review and any attached files.

Reviewer #1: No

---

## [Author Response · Author response to Decision Letter 0]

3 Aug 2022

* This response is contained in the document uploaded "Response to Reviewers". 

Reviewer #1: In their manuscript „An open-source, low-cost voluntary running activity tracking tool for in vivo rodent studies “, the authors describe a “simple” solution to a cost-effective method for tracking activity data in a multi-cage setup. They use an Arduino microcontroller to process the data and Matlab/R to evaluate the sensor robustness and statistics. An in vivo caloric restriction experiment is selected as a use case to demonstrate the method.

First, I would like to say that I am very familiar with commercial systems that can do the same but are rather pricy. Therefore, I welcome this “hands-on” method to establish a validated and low-cost solution to obtain multiple time-resolved activity patterns. Furthermore, getting total wheel-running counts over time (e.g., overnight) is trivial. Therefore, the innovative focus of this study lies in a) cost-effectiveness, b) the ability to run n cages in parallel, and c) the time resolution of activity patterns. I find it a bit sad that the analytical focus of the analysis was not more on c), but I also understand that the main aim of the manuscript was to present the validated setup.

The points a-c are addressed in the manuscript. I want to thank the authors for their work and encourage them to continue this kind of work and, e.g., develop more measures for severity assessment that can be implanted this easily.

The manuscript is well-written, good to understand, and straightforward. The authors also address critical behavior-related activity issues like housing conditions, increased anxiety, and food-related stress.

Thank you very much for valuing our study and the idea presented here of circumventing cost obstacles to track rodent activity in their home cage. We agree with the reviewer that we did not emphasize enough the real time tracking capability of this device. We have added a measure of velocity to Figure 3A to show the time resolution of both distance and velocity over the course of the study, to demonstrate that subtle changes and trends throughout the light-dark cycle can be detected, and have expanded the discussion regarding this aspect of our work page 9, line 185.

However, some minor things need to be addressed before I can recommend the manuscript for publication.

a) Since this is an animal study (at least the caloric restriction experiment): did the authors include the ARRIVE guidelines with the manuscript?

Thank you for pointing us to this useful resource, we appreciate it. We have downloaded and checked all the items in the E10 guideline for our study to make sure this paper will be easily reproducible and useful to the scientific community. We have uploaded the E10 checklist to the GitHub repository for this project to make sure it is openly available with the rest of our data and manuscript.

b) In the caloric restriction experiment, there is no hypothesis. However, the authors mention that they measured the difference in distance between “baseline and caloric restriction measurements” and that this was statistically significant.

o I cannot see an initial hypothesis and how the effect is (potentially) biologically meaningful

Thank you for this comment. We have added clearly our hypothesis in line 191 on page 9, which is that caloric restriction in mice will exacerbate foraging behavior which will translate on the motricity tracker by a higher recorded traveled distance. 

o I guess that there was no power analysis done before the experiment. Therefore, we cannot know whether the result is sufficiently powered or meaningful. If the authors have done an a priori power analysis, they should include it (with their hypothesis). 

 If they haven’t, they should explain their level of significance threshold.

 The term “statistically significant” is not self-sufficient as a result. Without an effect or hypothesis, this statement is meaningless.

o A type-1 error or α-error of p<= 0.05 is the threshold for “statistical significance” in general science. However, the results report “p=0.0876” (line 170) above this threshold. Therefore, the result is NOT significant. Why was it termed “significant” in the text?

o However, scientists can (when there is a good reason) change that threshold, e.g., to p=0.1 (e.g., if they followed Fisher’s definition of the p-value). In this case, the result would be significant. But without a hypothesis, this is again meaningless, and the authors also give no reason why the level should be larger than the commonly accepted threshold.

Given that we were testing this for the first time and were using this experiment as an assessment of the efficacy of the tracker system, we did not do a power analysis beforehand. If we approximate that the data follow a normal distribution, we determined via power analysis using the pwr package in R that we needed 10 samples, or cages (paired; assessed both prior to and following the caloric restriction phase) if we wanted to be able to find a significant difference between the two groups based on a power level of 0.8 and a significance level of 0.05. This result relates to the reviewers comment about the significance of the results: we found the difference to not be significant (i.e. p value < 0.05) but found a trend (we have edited our manuscript page 10, line 195 to reflect this). We have now added these points in the discussion page. We believe however that we have demonstrated the benefits and robustness of our tool as used in an in vivo scenario when compared to the mechanical wheel.

o Please provide context for the reported p-value and why this should be significant and/or adjust the reporting of the result.

 We thank the reviewer for pointing out this oversight. We have edited the results section as well as indicated above to reflect the non significance of the results and kept a threshold at 0.05 (as indicted above in the power analysis)

c) Typo in line 170: “varFiation” (variation?)

Thank you for noticing this, we have fixed this typo in the text.

d) The authors analyzed the data “through calculation of the empirical cumulative distribution using 1,000 label permutations, followed by a one sided tailed test”.

o This needs a better explanation: What kind of test did the authors use and why?

We agree, this does require a more complete explanation. We’ve added the following to the methods section, line 149: 

We used a permutation test on the means of the differences between each cage before/after caloric restriction distance traveled. In more detail: we measured the distance each cage (n = 8, with 3 mice per cage) ran for 3 days, 10 hours a day and summed that distance per cage (‘before distance’). Then we measured the distance each cage ran after caloric restriction for 3 days, 10 hours a day, and summed that distance (‘after distance’). We generated 1000 permutations by randomly shuffling each cages’ before/after distance, and took the mean across cages. These means produce the null distribution. We then measured the actual mean of before/after distances, and calculated the area under the curve that is more extreme than the actual measured mean value. We chose this test over a t test to account for the non-normal distribution, and over a Wilcoxon test as we wished to test the difference in the means in a non-parametric way (rather than the rank sum).

o I understand a permutation test and that this test can, e.g., be a t-test that is permuted on the ECDF data differences. Was this a t-test? The term “one sided tailed test“ is incorrect. Usually, this is called a one-tailed xy-test. Sided and tailed means the same here.

Thank you for pointing this out, it was an oversight, here we performed a permutation test. We have removed the term “one sided tailed test” from the text. 

o And, why was a one-tailed test chosen and not a two-tailed (this information should be given in the missing hypothesis as mentioned above)? The one-tailed design hints that the authors at least expected a lower/higher development in one of the groups, other than a general difference (two-tailed).

As explained above, we did not use a t test for this study, it was an oversight and we appreciate the reviewer’s comment regarding this mistake. 

o ECDF functions can also be analyzed with a Kolmogorov-Smirnov test; Was it a KS-test?

We are not completely clear about the question, but we hope that we have answered the reviewer's question in our previous answer.

---

## [Decision Letter · Decision Letter 1]

17 Aug 2022

An open-source, low-cost voluntary running activity tracking tool for in vivo rodent studies

PONE-D-21-39460R1

Dear Dr. Deitzler,

We’re pleased to inform you that your manuscript has been judged scientifically suitable for publication and will be formally accepted for publication once it meets all outstanding technical requirements.

Kind regards,

Prof. Dr. Dragan Hrncic, MD, MSc, PhD 

Academic Editor

PLOS ONE

Additional Editor Comments (optional):

Up date the repository.

Reviewers' comments:

Reviewer's Responses to Questions

**Comments to the Author**

1. If the authors have adequately addressed your comments raised in a previous round of review and you feel that this manuscript is now acceptable for publication, you may indicate that here to bypass the “Comments to the Author” section, enter your conflict of interest statement in the “Confidential to Editor” section, and submit your "Accept" recommendation.

Reviewer #1: All comments have been addressed

2. Is the manuscript technically sound, and do the data support the conclusions?

Reviewer #1: Yes

3. Has the statistical analysis been performed appropriately and rigorously? 

Reviewer #1: Yes

4. Have the authors made all data underlying the findings in their manuscript fully available?

Reviewer #1: Yes

5. Is the manuscript presented in an intelligible fashion and written in standard English?

Reviewer #1: Yes

6. Review Comments to the Author

Reviewer #1: Please upload the E10 checklist to the GitHub repo as stated. The last change in the repository was on " Oct 6, 2021".

7. PLOS authors have the option to publish the peer review history of their article (what does this mean?). If published, this will include your full peer review and any attached files.

Reviewer #1: No

---

## [Editor Report · Acceptance letter]

30 Aug 2022

PONE-D-21-39460R1 

An open-source, low-cost voluntary running activity tracking tool for *in vivo* rodent studies 

Dear Dr. Deitzler:

I'm pleased to inform you that your manuscript has been deemed suitable for publication in PLOS ONE. Congratulations! Your manuscript is now with our production department. 

Kind regards, 

on behalf of

Professor Dragan Hrncic 

Academic Editor

PLOS ONE